# Frequency, Treatment and Outcome of Immune-Related Toxicities in Patients with Immune-Checkpoint Inhibitors for Advanced Melanoma: Results from an Institutional Database Analysis

**DOI:** 10.3390/cancers13122931

**Published:** 2021-06-11

**Authors:** Florentia Dimitriou, Ramon Staeger, Melike Ak, Matias Maissen, Ken Kudura, Marjam J. Barysch, Mitchell P. Levesque, Phil F. Cheng, Reinhard Dummer, Joanna Mangana

**Affiliations:** 1Department of Dermatology, University Hospital of Zurich, 8091 Zurich, Switzerland; florentia.dimitriou@usz.ch (F.D.); ramon.staeger@usz.ch (R.S.); melike.ak@usz.ch (M.A.); marjam.barysch-bonderer@usz.ch (M.J.B.); mitchell.levesque@usz.ch (M.P.L.); phil.cheng@usz.ch (P.F.C.); johanna.mangana@usz.ch (J.M.); 2Faculty of Medicine, University of Zurich, 8006 Zurich, Switzerland; matias.maissen@uzh.ch; 3Department of Nuclear Medicine, University Hospital Zurich, 8091 Zurich, Switzerland; ken.kudura@usz.ch

**Keywords:** immune-related adverse events, melanoma, immunotherapy, infliximab, tocilizumab

## Abstract

**Simple Summary:**

In this study, we investigated the impact of immune-related adverse events (irAEs) on the survival of advanced melanoma patients treated with immune-checkpoint inhibitors, as well as the effect of corticosteroids and other immune-modulators on clinical outcome. We summarized the kinetics, onset, and outcome of immune-related adverse events (irAEs) in both adjuvant and non-adjuvant settings and we correlated their onset with disease outcome.

**Abstract:**

Immune checkpoint inhibitors (ICIs) can induce immune-related adverse events (irAEs), which may result in treatment discontinuation. We sought to describe the onset, frequency, and kinetics of irAEs in melanoma patients in a real-life setting and to further investigate the prognostic role of irAEs in treatment outcomes. In this retrospective single-center cohort study, we included 249 melanoma patients. Onset, grade, and resolution of irAEs and their treatment were analyzed. A total of 191 (74.6%) patients in the non-adjuvant and 65 (25.3%) in the adjuvant treatment setting were identified. In the non-adjuvant setting, 29 patients (59.2%) with anti-CTLA4, 43 (58.1%) with anti-PD1, and 54 (79.4%) with anti-PD1/anti-CTLA4 experienced some grade of irAE and these had an improved outcome. In the adjuvant setting, the frequency of irAEs was 84.6% in anti-CTLA4 and 63.5% in anti-PD1, but no correlation with disease relapse was observed. Patients with underlying autoimmune conditions have a risk of disease exacerbation. Immunomodulatory agents had no impact on treatment efficacy. IrAEs are correlated with increased treatment efficacy in the non-adjuvant setting. Application of steroids and immunomodulatory agents, such as anti-TNF-alpha or anti-IL6, did not affect ICI efficacy. These data support irAEs as possible prognostic markers for ICI treatment.

## 1. Introduction

Immune checkpoint inhibitors (ICIs) such as anti-PD1 (programmed cell death 1) and anti-CTLA4 (cytotoxic T lymphocyte antigen 4) antibodies have revolutionized the therapeutic landscape of metastatic melanoma and are approved as first-line therapies in the advanced as well as the adjuvant setting [1,2,3,4]. However, the resulting disinhibition of T-cell responses can lead to a spectrum of immune-related adverse events (irAEs) that may potentially involve any organ, but most commonly affect the skin, gastrointestinal tract, endocrine glands, liver, lung, and musculoskeletal system [5]. Immune-related AEs are usually manageable, yet in certain cases they can lead to ICI discontinuation or rarely, be fatal [6,7]. In patients treated with combined anti-CTLA4/anti-PD1 (ipilimumab and nivolumab), irAEs of any grade can be observed with a frequency of up to 90% and may lead to treatment discontinuation in up to 50% of patients [8]. Monotherapies have a better safety profile; interestingly, the frequency and severity of irAEs differ with a higher incidence and grade in anti-CTLA4 than in anti-PD1 regimens [1].

In the treatment of the vast majority of irAEs, corticosteroids are the standard of care. In cases of steroid-refractory irAEs like colitis, pneumonitis, arthritis, and hepatitis, other immunomodulatory treatments such as blocking antibodies against tumor necrosis factor alpha (TNF-α) or interleukin 6 (IL6) were shown to be effective [5,9,10,11]. It remains unclear whether these interventions have a negative impact on the antitumor immune response. A pooled analysis of data from the CheckMate 069 and CheckMate 067 clinical trials showed that corticosteroids do not appear to inhibit tumor response [12]. However, high-dose steroids for long durations during anti-PD1 therapy may be associated with poorer survival outcomes [13].

Currently, there are conflicting reports on the prognostic role of irAEs. In some solid tumors, mainly non-small-cell lung cancer (NSCLC), several retrospective studies clearly correlate the occurrence of irAEs under anti-PD1 with an improved response [14,15]; whereas in melanoma, this relationship is less clear [14,16,17,18].

The aim of this study was to describe the frequency and kinetics of irAEs in melanoma patients treated with ICIs both in unresectable/non-adjuvant stage III/IV and in the adjuvant setting. Furthermore, we investigated the prognostic impact of irAEs on survival and the effect of steroids and other immunomodulators on clinical outcome.

## 2. Materials and Methods

In this single-center retrospective cohort study, we searched the melanoma registry of the Department of Dermatology, University Hospital Zurich, Switzerland, which is comprised of 2766 patients. All adult patients diagnosed with any subtype of melanoma, who received ICI treatment in any line in the adjuvant or unresectable/non-adjuvant setting from February 2011 to May 2020 and with a follow-up period of at least 3 months were included. In patients with multiple ICI treatment lines, the treatment with a confirmed irAE was chosen; if more than one treatment line had confirmed irAEs, then we chose the first ICI treatment; in patients without any toxicities, the first ICI treatment was defined as baseline treatment. The demographical, clinical, and pathological features of eligible patients, including age, sex, primary melanoma origin, stage (according to the American Joint Committee on Cancer 8th edition), and tumor treatment were obtained from our institutional database. Moreover, the electronic medical records of included patients were reviewed for history of autoimmune or auto-inflammatory co-morbidities prior to treatment with ICIs. Date of onset, grade, irAE-directed treatment such as steroids and other immunosuppressive/-modulatory agents, and resolution of irAEs were retrieved. Grading of irAEs was based on the Common Terminology Criteria for Adverse Events (CTCAE), version 5.0 [19].

Overall survival (OS) was calculated from the date of therapy initiation to the last follow-up or death. Progression-free survival (PFS) was calculated from the date of therapy initiation to documented disease progression or last follow-up for non-progressed patients. Three-monthly positron emission tomography–computed tomography (PET-CT) or computed tomography (CT) scans according to the institution protocols were used for the assessment of treatment responses evaluated according to the RECIST criteria [20].

A total of 256 patients were subjected to analysis. Categorical variables were summarized as frequencies. To assess differences in categorical variables, the Fisher’s exact test was used. Continuous variables were summarized as median and range. For continuous variables, the Wilcoxon rank test was used. Survival differences were assessed with the log-rank test. A *p*-value of < 0.05 was considered as statistically significant. Time-to-event outcomes (duration of corticosteroids and immunosuppressive agents, time to resolution, OS, PFS) were visualized by Kaplan–Meier (KM) curves. All statistical analyses were performed by using R version 4.0.1.

## 3. Results

### 3.1. Demographics

A total of 256 patients, 191 (74.6%) in advanced/non-adjuvant and 65 (25.3%) in adjuvant setting, with median age of 63 years (range 25–92) were identified (Table 1).

Of these, 170 patients (66.4%) experienced some grade (1–4, CTCAEv5) of toxicity. The majority of the primary tumors in the overall study cohort were cutaneous (*n* = 201, 79%) and unknown primary (*n* = 24, 9.4%), and most of the patients had stage IV disease (*n* = 187, 73%). The vast majority of the patients (*n* = 186, 73%) were treatment naïve, whereas 27 patients (11%) had previously immunotherapy and 24 patients (9.4%) targeted therapy. Systemic treatments included anti-PD1/anti-CTLA4 in 68 (27%) patients, single-agent anti-PD1 in 126 (49%) patients and anti-CTLA4 in 62 (24%) patients. Median follow up (FU) was 27 months (range 3–108) and at the end of the observation period, 92 (36%) patients died due to melanoma, one (0.4%) patient died from adverse events and seven (3%) patients died of other causes, while the rest of 156 (61%) patients were still alive.

### 3.2. Adverse Events

In the overall study population, treatment-related toxicities were reported in 11 (84.6%) patients with anti-CTLA4 and in 33 patients (63.5%) patients with anti-PD1 in the adjuvant setting. A total of 29 patients (59.2%) with anti-CTLA4, 43 (58.1%) patients with anti-PD1 and 54 (79.4%) patients with anti-PD1/anti-CTLA4 in the non-adjuvant setting had treatment-related toxicities (Figure 1A). Median time to onset of irAEs was 52 days (range 0–1328). Overall, the most common treatment-related toxicities included thyroiditis (29.4%), colitis (27.6%), rash (24.7%), hepatitis (18.2%), arthritis (13.5%), hypophysitis (12.9%), pneumonitis (8.8%), vitiligo-like depigmentation (8.8%) and pancreatitis (7.6%). Other adverse events of special interest included myocarditis (3.5%), cytokine release syndrome (CRS, 1.8%), type 1 diabetes (1.2%), uveitis (1.2%), encephalitis (0.6%), meningitis (0.6%), myositis (0.6%) and nephritis (0.6%). In the non-adjuvant setting, combined immunotherapy was complicated with severe (grade ≥ 3) toxicities in 58.8% of the cases, compared to 22.4% and 13.3% in anti-CTLA4 and anti-PD1 agents, respectively. In the adjuvant setting, grade 3 and higher toxicities were reported in 38.4% and 13.5% of patients treated with anti-CTLA4 and anti-PD1, respectively (Figure 1B).

Late-onset toxicities, i.e., toxicities that were observed after treatment discontinuation, were reported in nine patients, with median time to onset 132 days (range 100–334) after treatment discontinuation and included most frequently endocrinological toxicities (*n* = 4, including hypophysitis, thyroiditis and type 1 diabetes), skin toxicities (*n* = 3), pneumonitis (*n* = 1) and inflammatory/CRS symptoms (*n* = 1). When stratified according to the treatment regimen, anti-PD1 treatment was complicated more frequently with immune-related arthritis, pneumonitis, myocarditis, thyroiditis and hematologic toxicities, whereas anti-CTLA4-driven toxicities included colitis, hypophysitis and skin rash; types of toxicities and kinetics according to treatment are summarized in Figure 2.

### 3.3. Adverse Events in Non-Adjuvant/Unresectable Setting

In the non-adjuvant/unresectable setting (AJCCv8 Stage III-IV), treatment-related toxicities were reported in 126 patients (65.6%), of which 45% experienced ≥2 toxicities (range 2–7) (Table 1). A total of 74 (39%), 68 (36%), and 49 (26%) patients were treated with anti-PD1, anti-CTLA4/anti-PD1 and anti-CTLA4 respectively. Treatment-related toxicities led to treatment discontinuation in 25 patients (20%), mostly in the combined immunotherapy group. Median time to toxicity onset was 43.5 days (range 0–1328); 31.5 days for anti-PD1/anti-CTLA4, 44 days for anti-CTLA4 and 97 days for anti-PD1 (Figure 1C). The most frequent toxicities for combined immunotherapy included immune-related thyroiditis (30.9%), hepatitis (29.4%) and colitis (26.5%). In the monotherapy regimens, anti-PD1 treatment was mostly associated with skin (vitiligo-like depigmentation 12.2%, rash 9.5%) and endocrinological (thyroiditis 19%, hypophysitis 9.5%) toxicities, whereas colitis (34.7%) and skin rash (22.4%) were commonly observed in single-agent anti-CTLA4 (Figure 3A).

Treatment discontinuation was most commonly related to combined immunotherapy, with similar patterns as in the general study population. Of note, immune-related arthritis led to treatment discontinuation in 8.8% of patients with combined immunotherapy, as well as 9.5% in single-agent anti-PD1. Similar to combination treatment, colitis was the most frequent event that led to treatment discontinuation in anti-CTLA4 alone (18.4% and 14.7%, respectively).

### 3.4. Adverse Events in Adjuvant Setting

Out of the 65 patients that were treated in the adjuvant setting (AJCCv8, Stage II–IV), 44 (67.7%) patients presented with treatment-related adverse events, prompting treatment discontinuation in 10 (23%) patients (Table 1). The selected treatment regimen was anti-PD1 alone in 33 (75%) patients, whereas 11 (25%) patients were treated with adjuvant anti-CTLA4, in a reduced adjuvant dose (3mg/kg Q3W). Sixteen (36.4%) patients had multiple toxicities (range 2–5). Median time to toxicity onset was 89.5 days (range 14–546); 115 days for anti-PD1 and 55 days for anti-CTLA4 (Figure 1C). In the single-agent anti-PD1, the toxicity pattern was similar to the non-adjuvant setting, with increasing frequency of hepatitis (9.6%), arthritis (9.6%) and colitis (7.7%) (Figure 3B).

Nevertheless, and compared to the non-adjuvant group, toxicities with long-term effects, or potentially fatal effects were more commonly observed and included endocrinological toxicities (thyroiditis 15.4%, hypophysitis 11.5%), myocarditis (5.7%), meningitis (1.9%) and encephalitis (1.9%). Immune-related colitis (5.8%), encephalitis (1.9%), meningitis (1.9%), myocarditis (1.9%) and pneumonitis (1.9%) were the grade 3–4 toxicities that led to treatment discontinuation in this group.

Similar to the non-adjuvant group, in the single-agent anti-CTLA4, colitis (30.8%) and hypophysitis (30.8%) were frequent treatment-related toxicities that were associated with severe, grade 3–4 adverse events in 23.1% and 15.4% of the cases, respectively.

### 3.5. Immunomodulatory Agents

Out of the 292 irAEs from 170 patients, 74 events were treated with steroids alone, 28 with steroids and an immunomodulatory treatment, including either anti-TNFα (*n* = 25) or anti-IL6 (*n* = 2) or both (*n* = 1). Patients received for six events anti-TNFα alone, and 184 received neither steroids nor an immunomodulatory treatment (Figure 3C). Within the 34 cases that received an immunomodulatory agent, the vast majority (23/34) of the toxicities were associated with anti-CTLA4/anti-PD1 and included colitis grade ≥2 (*n* = 13), hepatitis grade ≥3 (*n* = 5), arthritis grade ≥2 (*n* = 3), gastritis grade 3 (*n* = 1) and pneumonitis grade 2 (*n* = 1). Other toxicities included arthritis grade 3 (*n* = 2) and colitis grade 2 (*n* = 1), associated with single-agent anti-PD1 and colitis grade ≥2 (*n* = 8), associated with anti-CTLA4 (Figure 3D). Anti-TNF-α was preferred as the immunomodulatory agent in most colitis cases (21/22) and treatment benefit was observed in 20/21 cases; one patient with grade 3 colitis was treated with first-line anti-TNF-α as a steroid-sparing option with toxicity resolution. Of note, no benefit was observed in two cases of grade 3 hepatitis and grade 3 colitis treated with anti-TNF-α, thus requiring further immunosuppression. Anti-IL6 was administered in one case of pneumonitis grade 2 and colitis grade 3, with toxicity resolution. A patient with colitis grade 3, who did not respond to either steroids or anti-TNF-α, was treated with anti-IL6, with event resolution.

### 3.6. Patients with Underlying Autoimmune/Auto-Inflammatory Comorbidities

Twenty (7.8%) patients with underlying autoimmune or auto-inflammatory conditions at baseline, including inflammatory bowel disease (IBD, *n* = 3), type 1 diabetes (*n* = 3), Bechterew’s disease (*n* = 1), multiple sclerosis (MS, *n* = 1), rheumatoid arthritis (RA, *n* = 6), chronic thyroiditis (*n* = 4) and sarcoidosis (*n* = 2) were identified (Table 1). At the time of immunotherapy initiation, seven (35%) patients had an inactive underlying disease, and five (25%) required symptomatic treatment or substitution only. Low-dose steroids or systemic immunosuppressive treatment was present in six (30%) cases. One patient with MS was on fingolimod treatment, which was discontinued prior to immunotherapy (anti-PD1/anti-CTLA4), without MS exacerbation during treatment. Notably, prophylactic treatment with anti-TNF-α was administered in a patient with inactive IBD, subsequently treated with anti-PD1, without an IBD flare.

The majority of the patients received anti-PD1 alone (*n* = 11); nevertheless, six patients were treated with anti-PD1/anti-CTLA4 and three with anti-CTLA4. Five (25%) patients tolerated the systemic treatment well, with neither treatment-related toxicities nor disease flare, whereas fifteen (75%) patients, four from the adjuvant and eleven from the advanced/unresectable setting, developed treatment-related toxicities and seven (35%) patients had exacerbation of their underlying disease. Disease flare (*n* = 4) of inactive conditions or worsening (*n* = 3) was noted in rheumatoid arthritis (*n* = 2), chronic thyroiditis (*n* = 3), IBD (*n* = 1) and type 1 diabetes (*n* = 1).

### 3.7. Efficacy

Patients that presented with irAEs in the adjuvant setting showed no superior prognosis with regard to relapse-free survival (RFS) compared to the patients without irAEs, in the different treatment regimens (*p* = 0.43 for anti-CTLA4 and *p* = 0.65 for anti-PD1, Figure 4A). Overall survival (OS) was similar between the two groups (Figure 4B), but mOS was not reached. Univariate Cox regression analysis for the clinical features (age, sex and presence of irAEs) showed no significant results for RFS and OS (Appendix A).

In the non-adjuvant setting, patients with irAEs presented with more favorable PFS compared to non-irAEs in all treatment regimens (*p* = 0.0036 for anti-CTLA4, *p* = 0.0022 for anti-PD1 and *p* < 0.0001 for anti-CTLA4/anti-PD1, (Figure 4C), which was also translated to survival benefit on OS rates except for anti-PD1 (*p* = 0.22, Figure 4D). Univariate Cox regression analysis for the clinical features (age, sex, elevated LDH at treatment start and presence of irAEs and brain metastases) showed that presence of irAEs was associated with prolonged PFS in anti-CTLA4 (HR 0.34, *p* = 0.007), anti-PD1 (HR 0.32, *p* = 0.002) and anti-PD1/anti-CTLA4 (HR 0.08, *p* < 0.001) (Appendix A). Male sex negatively influenced PFS in anti-CTLA4 treatment (HR 2.84, *p* = 0.024). Elevated LDH at treatment start was associated with poor OS in anti-CTLA4 (HR 8.04, *p* < 0.001), anti-PD1 (HR 2.67, *p* = 0.034) and anti-PD1/anti-CTLA4 (HR 0.16, *p* < 0.001). The presence of irAEs was significant for prolonged OS in anti-CTLA4 (HR 0.34, *p* = 0.026) and anti-PD1/anti-CTLA4 (HR 0.16, *p* < 0.001).

Patients that experienced an irAE were assessed for whether no treatment or treatment with steroids with or without an immunomodulatory agent affected PFS and OS. Of note, steroids and other anti-inflammatory drugs had no significant impact on progression-free survival or overall survival in the adjuvant and non-adjuvant setting (PFS and OS, Figure 5A–D).

## 4. Discussion

Although ICIs are currently in widespread use in oncology and their efficacy in melanoma patients is unquestionable, there is contradictory evidence on the correlation of irAEs with clinical outcome. In this retrospective study, patients with irAEs showed improved PFS in all unresectable/non-adjuvant treatment regimens, compared to patients without irAEs. In combined immunotherapy with anti-PD1/anti-CTLA4, the presence of irAEs was associated with an OS benefit as well. As such, irAEs may represent an indicator of treatment efficacy in these patients. Of note, in the adjuvant setting, no correlation of irAEs and relapse-free or OS was observed. Organ-specific irAEs differed between the two agents; anti-PD1 treatment was complicated more frequently with arthritis, pneumonitis, myocarditis, thyroiditis and hematologic toxicities, whereas anti-CTLA4-driven toxicities included colitis, hypophysitis and skin rash. Treatment of irAEs with steroids and other immunomodulatory agents, such as anti-TNFα and anti-IL6, had no impact on treatment efficacy.

Our results are in line with several retrospective studies suggesting significant benefit towards tumor control and survival in immunotherapy-treated patients that experience any kind of irAE, with the majority of reports coming from non-small-cell lung cancer field (NSCLC) [14,15,21,22]. However, for melanoma, the association between toxicity and efficacy is not as linear as observed in other cancer types. The majority of published studies have indicated either a confirmed PFS or OS benefit for dermatologic toxicities (especially for hypopigmentation) or no benefit at all; a retrospective study of melanoma patients treated with anti-PD1 reported OS benefit in patients with irAEs but in the multivariate analysis only ≥ grade 3 toxicities remained a significant confounder for OS [23,24]. A pooled analysis of the Keynote 001, 002 and 006 clinical trials in advanced melanoma patients treated with pembrolizumab demonstrated no efficacy difference in patients with irAEs compared to no irAE at the landmark analysis of 21 weeks. Still, the analysis was conducted only in 29.5% of the total population (463/1567 patients) including only those that did not progress until week 21, therefore excluding both patients with innate early resistance or those with late toxicities [18].

Notably, the frequency and type of irAE differed significantly in the adjuvant setting. This was particularly prominent in anti-PD1 treatment and in all grades of irAEs, but frequency of ≥ grade 3 irAEs was similar between the adjuvant and non-adjuvant setting. This observation is pertinent for risk/benefit assessment in adjuvant treatment, as many patients may already be cured by surgery alone. Furthermore, we observed a higher incidence of toxicities with long-term effects, or potentially fatal toxicities in the adjuvant setting with an increased frequency of myocarditis (6.7%), meningitis (2.2%), encephalitis (2.2%) and endocrinological toxicities (thyroiditis 17.8% and hypophysitis 13.3%). In fact, irAEs with long-term or potentially fatal effects were more commonly observed than in the non-adjuvant setting and this risk should not be neglected in adjuvant treatment decision in resected melanoma [25]. The difference in frequency of irAEs might be attributed to the treatment exposure time and the patient’s intrinsic factors, as well as immune status in stage III compared to advanced, unresectable melanoma [26]. In contrast to a recent report of the EORTC-1325/Keynote-054 study in patients with resected Stage III melanoma, where the presence of endocrine and skin toxicities in the pembrolizumab arm was correlated with improved RFS rates, our study did not confirm this superiority in risk of progression in the adjuvant setting [17]. However, the small number of patients in this setting precludes any statistical conclusion. Interestingly and besides the appearance of vitiligo-like depigmentation under interferons [27], no other reports with anti-PD1 or anti-CTLA4 monotherapy have been published so far confirming the prognostic value of irAEs in the adjuvant setting either prospectively or retrospectively.

In our cohort, 66.4% experienced at least one irAE; 79% with anti-PD1/anti-CTLA4, 60% with anti-PD1 and 64% with anti-CTLA4 alone, which is in the expected range, compared to data from randomized clinical trials. Notably, median time to toxicity onset was longer in the adjuvant setting versus non-adjuvant (89.5 versus 43.5 days). This is probably treatment-regimen-associated since more than 2/3 of the patients in the non-adjuvant setting received an anti-CTLA4-directed treatment, either alone or in combination with anti-PD1. Besides, nivolumab was initially tested at 3 mg/kg 2-weekly in the CheckMate-238 trial [4] and nivolumab/ipilimumab are administered at a dose of 1 and 3 mg/kg 3-weekly at the induction phase. Nivolumab, which is the treatment regimen preferred in the adjuvant setting, is now also approved and administered as a flat dose of 480 mg 4-weekly. While it is likely that such alterations in dosing and schedule do not impact efficacy or toxicity, the optimal treatment duration is not known and data from alternative dosing schedules from anti-CTLA4 should be thoroughly reviewed. In the adjuvant setting, ipilimumab is approved at a dose of 10 mg/kg, but its use is limited due to high toxicity. As such, we applied adjuvant ipilimumab in the conventional dose of 3 mg/kg 3-weekly, with significantly improved tolerability [28]. Although in the Checkmate-511 trial the induction dose of anti-CTLA4 does not seem to affect efficacy [29], in the Checkmate-915 trial, low-dose ipilimumab combined with nivolumab was not superior to nivolumab alone [30].

Our data suggest that steroids for irAEs or other immunomodulating agents for steroid-refractory toxicities do not have a direct impact on prognosis. Similar results have been reported in other studies and in a pooled analysis of pembrolizumab with patients that received steroids for irAEs experiencing similar efficacy outcomes with those without [12,18,31]. However, the impact of duration of the immunosuppressive treatment on disease outcome was not addressed in this study. In a secondary analysis of the EORTC-1325/Keynote-054 study, treatment effect of adjuvant pembrolizumab after 30 days of systemic steroid use appeared to be lower than without steroid use or by day 30 of systemic steroid use [17]. In another recent multicenter observational study on concomitant medications and outcomes in patients with solid tumors treated with ICIs, patients with steroids at baseline for cancer-related indications had significantly worse PFS and OS than those without or those with steroids for non-cancer-related indications [32]. Nevertheless, the authors did not include patients with irAEs requiring steroids in the non-cancer-related indications group. Taken all together, we assume that steroids given for irAEs for a short time or those given for symptom control have different effects on survival and outcome with the first neutralizing the detrimental effect by the positive effect of immune-mediated toxicity itself. Approximately 20% of steroid-refractory irAEs required additional treatment, with the majority of those having received TNF-α inhibitors. Rapid resolution was observed in almost all cases with colitis. The relatively high rate of TNF-α or IL-6 inhibitors in our cohort is a result of our clinical judgment that early treatment or treatment in lower irAEs grades (e.g., grade 2) is preferable to prolonged treatment with high-dose corticosteroids.

A minority in our study cohort had underlying autoimmune toxicities, with only 35% of those experiencing a flare of their disease and 75% of those experiencing any kind of irAE. A previous study has reported a similar incidence of autoimmune disease exacerbation in 52 melanoma patients treated with anti-PD1 monotherapy [33]. In the same study, no patients with previous gastrointestinal disease flared upon exposure to ICIs: however, one patient in our study with inactive IBD started prophylactic treatment with a TNF-α inhibitor upon initiation of ICIs. This probably suggests that the pathogenesis of autoimmunity is heterogeneous with many disorders not relying on PD1 or CTLA4 pathway.

Noteworthy, 5% (*n* = 9) of our patients experienced late-onset irAEs, defined as an irAE that occurred after treatment discontinuation, with endocrinological toxicities, skin toxicities and pneumonitis being the most frequent ones. These findings are comparable with a recent multicenter retrospective analysis reporting an estimated incidence of 5.7% of late onset irAEs and an identical pattern [34]. Although the incidence is low, late-onset irAEs can be difficult to manage and are definitely affecting patient’s quality of life.

Our study has a number of limitations, including the retrospective, single-center design and low number of patients in some subgroups especially in the adjuvant setting, those with prior autoimmune diseases and those receiving steroids or other immunomodulating agents for irAEs. In addition, we must consider that since the cohort was not collected prospectively, under-reporting of mild toxicities cannot be excluded. On the other hand, it represents a relatively large homogenous cohort of exclusively melanoma patients treated with ICIs beyond clinical trials.

## 5. Conclusions

In conclusion, the appearance of irAEs in the non-adjuvant setting is undoubtedly associated with significant outcome benefit. We confirm previous reports on the low number of autoimmune disease flares with ICIs and support the non-significant impact of steroids and other immunomodulating agents, when administered early and for a short time, for irAE management on clinical outcome. Further elucidation of the predictive and prognostic role of irAEs in the prospective setting is warranted.

## Figures and Tables

**Figure 1 cancers-13-02931-f001:**
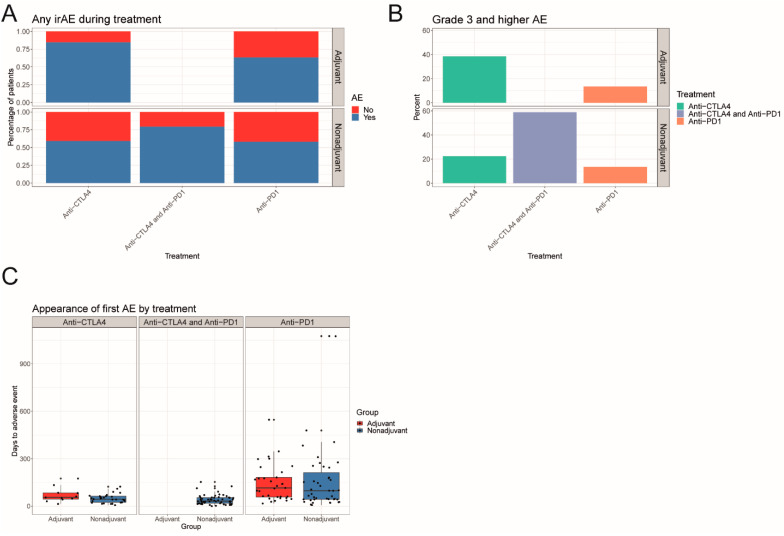
Frequency and time of appearance of irAEs in the adjuvant and non-adjuvant setting. (**A**) Barplot representing the frequencies of any irAE by treatment faceted by the adjuvant and non-adjuvant setting. (**B**) Barplot representing the frequencies of grade 3 and higher irAE by treatment faceted by the adjuvant and non-adjuvant setting. (**C**) Boxplots representing the appearance of the first irAE by treatment in the adjuvant and non-adjuvant setting.

**Figure 2 cancers-13-02931-f002:**
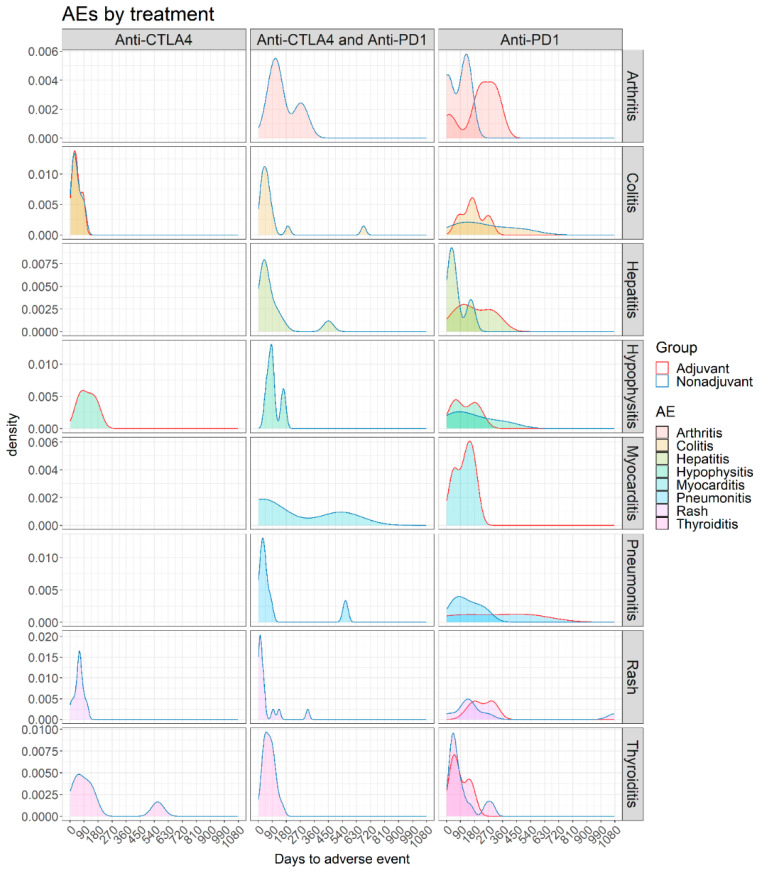
Density plots representing the time of appearance of arthritis, colitis, hepatitis, hypophysitis, myocarditis, pneumonitis, rash and thyroiditis in adjuvant and non-adjuvant patients treated with anti-CTLA4, anti-PD1, and anti-CTLA4 with anti-PD1.

**Figure 3 cancers-13-02931-f003:**
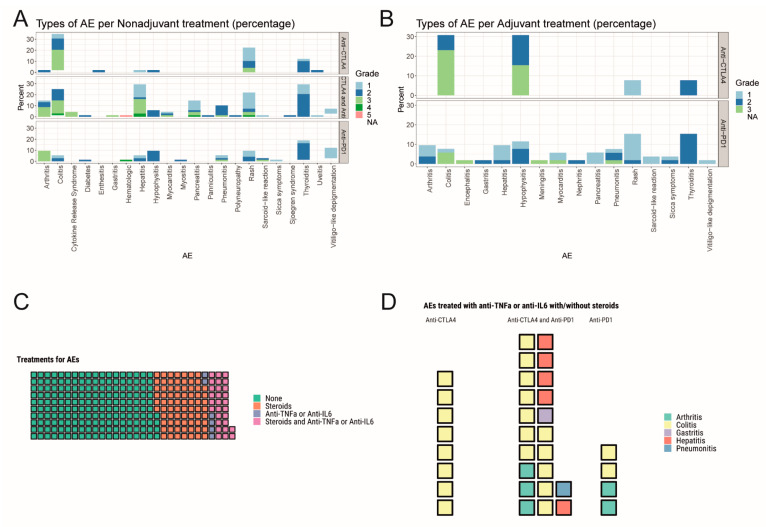
Types/grades of irAEs and their treatments in the adjuvant and non-adjuvant setting. (**A**) Stacked barplots representing the frequency of each time of irAE for anti-CTLA4, anti-PD1, and anti-CTLA4 with anti-PD1 in the non-adjuvant setting. (**B**) Stacked barplots representing the frequency of each time of irAE for anti-CTLA4 and anti-PD1 in the adjuvant setting. (**C**) Waffle plot showing the 292 irAEs untreated, treated with steroids, treated with anti-TNFα or anti-IL6, and treated with steroids and anti-TNFα or anti-IL6. (**D**) Waffle plot showing the different irAEs treated with anti-TNFα or anti-IL6 with and without steroids for anti-CTLA4, anti-PD1, and anti-CTLA4 with anti-PD1.

**Figure 4 cancers-13-02931-f004:**
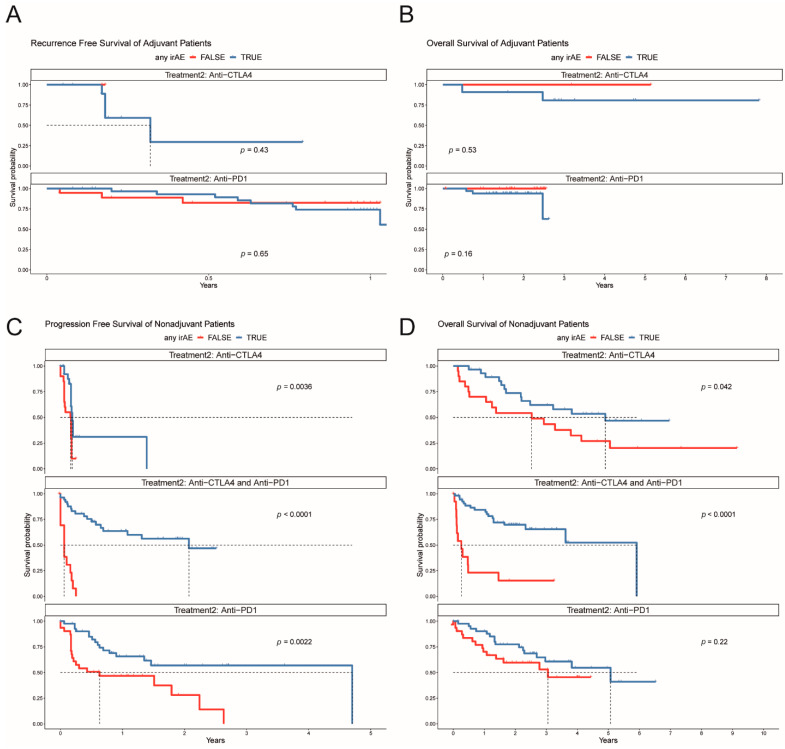
Kaplan–Meier plots for the adjuvant and non-adjuvant setting stratified by the appearance of irAE. (**A**) Recurrence-free survival for anti-CTLA4 and anti-PD1 in the adjuvant setting. (**B**) Overall survival for anti-CTLA4 and anti-PD1 in the adjuvant setting. (**C**) Progression-free survival for anti-CTLA4, anti-PD1, and anti-CTLA4 with anti-PD1 in the non-adjuvant setting. (**D**) Overall survival for anti-CTLA4, anti-PD1, and anti-CTLA4 with anti-PD1 in the non-adjuvant setting.

**Figure 5 cancers-13-02931-f005:**
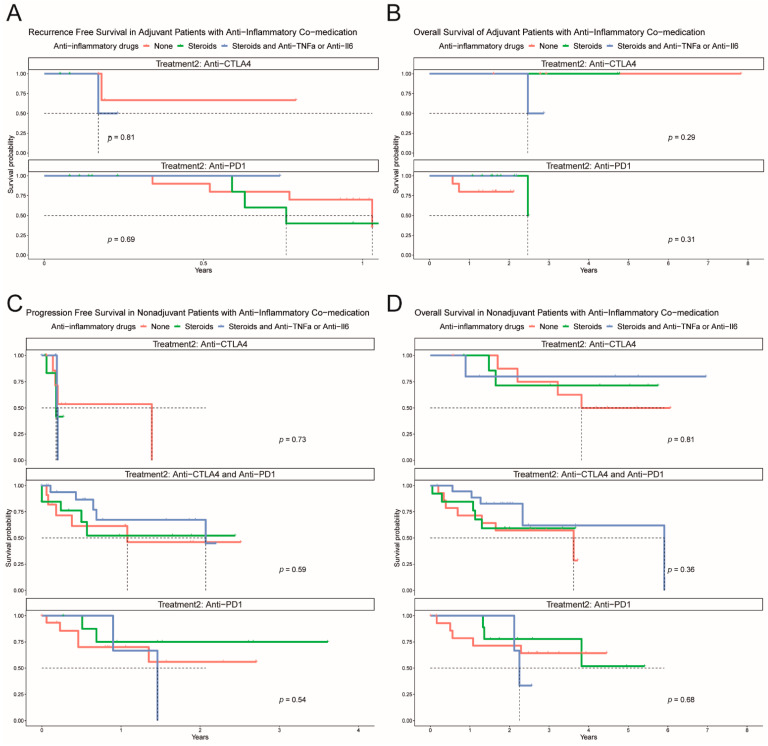
Kaplan–Meier plots for the adjuvant and non-adjuvant settings stratified by treatment for the irAE. (**A**) Recurrence-free survival in the adjuvant setting for anti-CTLA4 and anti-PD1. (**B**) Overall survival in the adjuvant setting for anti-CTLA4 and anti-PD1. (**C**) Progression-free survival in the non-adjuvant setting for anti-CTLA4, anti-PD1, and anti-CTLA4 with anti-PD1. (**D**) Overall survival in the non-adjuvant setting for anti-CTLA4, anti-PD1, and anti-CTLA4 with anti-PD1.

**Table 1 cancers-13-02931-t001:** Patients’ characteristics in the overall population, non-adjuvant and adjuvant setting.

Characteristic	Overall Population, *n* = 256	Non-Adjuvant Setting, *n* = 191			Adjuvant Setting, *n* = 65		
		Patients with No Toxicities, *n* = 65	Patients with Toxicities, *n* = 126	*p*-Value ^1^	Patients with No Toxicities, *n* = 21	Patients with Toxicities, *n* = 44	*p*-Value ^1^
**Age at treatment start (range)**	63 (25–92)	68 (35–89)	61 (33–92)	0.07	63 (31–83)	55 (25–87)	0.2
**AJCCv8 Stage at treatment start, *n* (%)**				0.8			0.8
II	4 (1.6%)	-	-	-	2 (9.5%)	2 (4.5%)	
III	65 (25%)	4 (6.2%)	10 (7.9%)		16 (76%)	35 (80%)	
IV	187 (73%)	61 (94%)	116 (92%)		3 (14%)	7 (16%)	
**Melanoma subtype, *n* (%)**				0.3			>0.9
Cutaneous	201 (79%)	49 (75%)	97 (77%)		18 (86%)	37 (84%)	
Acral	3 (1.2%)	2 (3.1%)	-		-	1 (2.3%)	
Mucosal	14 (5.5%)	4 (6.2%)	7 (5.6%)		1 (4.8%)	2 (4.5%)	
Unknown primary	24 (9.4%)	7 (11%)	11 (8.7%)		2 (9.5%)	4 (9.1%)	
Uveal	14 (5.5%)	3 (4.6%)	11 (8.7%)		-	-	
**Autoimmune disease in medical history, *n* (%)**				0.7			0.4
Chronic thyroiditis	5 (2.0%)	2 (3.1%)	2 (1.6%)		1 (4.8%)	-	
Diabetes type 1	3 (1.2%)	1 (1.5%)	2 (1.6%)		-	-	
Inflammatory bowel disease	2 (0.8%)	-	1 (0.8%)		-	1 (2.3%)	
Rheumatoid arthritis	6 (2.3%)	-	3 (2.4%)		-	3 (6.8%)	
Multiple Sclerosis	1 (0.4%)	-	1 (0.8%)		-	-	
Bechterew’s disease	1 (0.4%)	1 (1.5%)	-		-	-	
Sarcoidosis	2 (0.8%)	-	2 (1.6%)		-	-	
**Presence of brain metastases at treatment start, *n* (%)**	48 (19%)	16 (25%)	32 (25%)	>0.9	-	-	
**Type of treatment, *n* (%)**				0.01			0.2
Anti-PD1/Anti-CTLA4	68 (27%)	14 (22%)	54 (43%)		-	-	
Anti-PD1	126 (49%)	31 (48%)	43 (34%)		19 (90%)	33 (75%)	
Anti-CTLA4	62 (24%)	20 (31%)	29 (23%)		2 (9.5%)	11 (25%)	
**Reason for treatment discontinuation, *n* (%)**							0.01
Toxicity	35 (13.7%)	-	25 (20%)		-	10 (23%)	
Disease progression/recurrence	112 (43.8%)	51 (79%)	46 (71%)		3 (14%)	12 (27%)	
Treatment completed	40 (15.6%)	1 (1.5%)	4 (3.2%)		16 (76%)	19 (43.1%)	
CR	34 (13.3%)	7 (11%)	27 (21%)		-	-	
Patient’s/Investigator’s choice	22 (8.6%)	5 (7.7%)	12 (9.5%)		2 (9.5%)	3 (6.8%)	
**Best Overall Response (BOR), *n* (%)**				<0.001			
CR	95 (40%)	8 (12%)	44 (37%)		-	-	
PR	36 (15%)	12 (19%)	24 (20%)		-	-	
SD	12 (5.0%)	2 (3.1%)	10 (8.4%)		-	-	
PD	95 (40%)	42 (66%)	41 (34%)		-	-	

^1^ Wilcoxon rank sum test; Fisher’s exact test; Pearson’s Chi-squared test; Abbreviations: CR; complete response, PR; partial response, SD; stable disease, PD; progressive disease.

## Data Availability

All data is available upon request. Written informed consent for the use and collection of data for the use in retrospective analysis was approved by the local ethics committee (MelProg Project KEK-ZH 2014-0193, KEK 2017-00494). The study was conducted in accordance with the Declaration of Helsinki.

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
