# Peer review of "Frequency, Treatment and Outcome of Immune-Related Toxicities in Patients with Immune-Checkpoint Inhibitors for Advanced Melanoma: Results from an Institutional Database Analysis"

_cancers, 2021, doi:10.3390/cancers13122931_

Round 1

Reviewer 1 Report

There are some small typographical errors but overall the manuscript is well written. 

The figures are difficult to read and interpret because the font is too small, therefore they don't help the reader understand the data. I suggest modifying these - i.e. showing less data in the figures and making each more legible

The sample size is quite small, particularly in the adjuvant setting, which is acknowledged. However, this does limit conclusions from that data set.

Is there a possibility of accessing more data from additional hospital data sets to increase the size of the sample and strengthen the conclusions.

Could Table 1 and Table 2 be combined and presented as landscape to the two cohorts could be viewed side by side?

Also is there a reason best overall response data is not included in the adjuvant table?

Could differences in patient characteristics in these two groups could be confounders for the relationship between AE and treatment efficacy in the two groups. For example, 100% of adjuvant patients had brain metastases compared to 25% in the non-adjuvant group? Could the authors comment?

Author Response

Reviewer 1

Comment 1: There are some small typographical errors but overall the manuscript is well written.

Response 1: Thank you very much for your review. Typographical errors have been corrected in the manuscript.

Comment 2: The figures are difficult to read and interpret because the front is too small, therefore they don’t help the reader understand the data. I suggest modifying these- i.e. showing less data in the figures and making each more legible.

Response 2: Thank you very much for your detailed feedback. We undertook some modifications to our figures. We agree that the text was too small. We modified all figures so that they are readable. We also omitted Figure 1E as well as figure 2E. All figures have been enlarged to make the captions of the graphs more legible. The changes in the figures can be tracked in the revised version of our manuscript.

Comment 3: The sample size is quite small, particularly in the adjuvant setting which is acknowledged. However, this does limit conclusions from that data set. Is there a possibility of accessing more data from additional hospital data sets to increase the size of the sample and strengthen the conclusions.

Response 3: A larger study cohort, especially of patients treated in the adjuvant setting would have helped to obtain more meaningful results, which could have strengthened our conclusions. In the discussion, we reported the low number as a limitation of our study: “Our study has a number of limitations including the retrospective, single-center design and low number of patients in some subgroups especially in the adjuvant setting”. More data could have been accessed if data sets from other melanoma centres in Switzerland were in included in the study. But, since our study was designed as a single-centre study, we focused on the melanoma registry of the Department of Dermatology, University Hospital of Zurich, Switzerland.

Comment 4: Could Table 1 and Table 2 be combined and presented as landscape to the two cohorts could be viewed side by side?

Response 4: Thank you for your comment. We agree that a combined table would make the results easier to capture. We merged Table 1 and Table 2, as requested (please refer to Table 1).

Comment 5:  Also is there a reason best overall response data is not included in the adjuvant table?

Response 5: Recurrence rates in the adjuvant setting can be derived from the “Reason of Treatment Discontinuation” in the revised Table 1.  

Comment 6: Could differences in patient characteristics in these two groups could be confounders for the relationship between AE and treatment efficacy in the two groups. For example, 100% of adjuvant patients had brain metastases compared to 25% in the non-adjuvant group? Could the authors comment?

Response 6: Univariate Cox-regression analysis of patients’ characteristics in unresectable/non-adjuvant patients for the different treatment regimens did not show any significances for OS and PFS for the presence of brain metastases. To highlight this important observation, we have added these figures as supplementary material in the manuscript: “Univariate Cox regression analysis for the clinical features (age, sex, elevated LDH at treatment start and presence of irAEs and brain metastases) showed that presence of irAEs was associated with prolonged PFS in anti-CTLA4 (HR 0.34, p=0.007), anti-PD1 (HR 0.32, p=0.002) and anti-PD1/anti-CTLA4 (HR 0.08, p<0.001) (Supplementary Figure s2, available online). Male sex negatively influenced PFS in anti-CTLA4 treatment (HR 2.84, p=0.024). Elevated LDH at treatment start was associated with poor OS in an-ti-CTLA4 (HR 8.04, p<0.001), anti-PD1 (HR 2.67, p=0.034) and anti-PD1/anti-CTLA4 (HR 0.16, p<0.001). Presence of irAEs was significant for prolonged OS in anti-CTLA4 (HR 0.34, p=0.026) and anti-PD1/anti-CTLA4 (HR 0.16, p<0.001).

Reviewer 2 Report

The study by Dimitriou et al detailed a systematic analysis of immune related adverse events (irAEs) in a retrospective cohort of melanoma patients treated with immune checkpoint inhibitors. The authors examined the occurrence, type and pattern of irAEs in this patient cohort and assessed its impact on treatment response. The findings reported here are interesting, albeit not novel, and similar studies have been performed recently. Moreover, there are some interesting observations that should be expanded on, that was neglected in the discussion. The manuscript can be further improved by addressing the following comments: 

  • There are minor errors in the abstract. For example, it should read “sought” instead of “ought” (page 1, line 24) and the following sentence after this is a fragment “A retrospective cohort study from an institutional database analysis”. (page 1, line 26).
  • The authors should comment on why there is higher frequency of irAEs in the adjuvant setting compared to the nonadjuvant.
  • The authors should comment on why irAEs are correlated with increased treatment efficacy in the non-adjuvant setting but not in the adjuvant. What biological, clinical or statistical factors could have contributed to this difference and is this expected?
  • Is there any association in irAEs between treatment naïve patients vs those who had previous immunotherapy vs those who had previous targeted therapy?
  • Many previous studies have reported that anti-CTLA4 is more toxic compared to anti-PD1. However, in this study, the percentage of patients experiencing treatment related toxicities with anti-CTLA4 (59.2%) was similar to those with anti-PD1 (58.1%) in the non-adjuvant setting. Can the authors comment on this difference?
  • In their study, the authors concluded that application of steroids and immunomodulatory agents did not affect ICI efficacy. However, from previous studies (reference 31 and 32), this may be dependent on the duration of anti-PD-1 therapy. Did the authors consider this in their cohort (ie is the analysis stratified by duration of immune checkpoint inhibitors)?
  • Figure 1D is a bit difficult to interpret, especially the x axis is not clear.
  • Figure 2C and 2D displayed as Waffle plots, although pretty to look at, is perhaps unnecessary and could be better summarised in a table format.

Author Response

Reviewer 2

The study by Dimitriou et al detailed a systematic analysis of immune related adverse events (irAEs) in a retrospective cohort of melanoma patients treated with immune checkpoint inhibitors. The authors examined the occurrence, type and pattern of irAEs in this patient cohort and assessed its impact on treatment response. The findings reported here are interesting, albeit not novel, and similar studies have been performed recently. Moreover, there are some interesting observations that should be expanded on, that was neglected in the discussion. The manuscript can be further improved by addressing the following comments: 

Comment 1: There are minor errors in the abstract. For example, it should read “sought” instead of “ought” (page 1, line 24) and the following sentence after this is a fragment “A retrospective cohort study from an institutional database analysis”. (page 1, line 26).

Response 1: Thank you for your response. Minor errors in the abstract and main body have been corrected.

Comment 2: The authors should comment on why there is higher frequency of irAEs in the adjuvant setting compared to the nonadjuvant.

Response 2: Thank you for your suggestion. The text was revised accordingly and we also underlined that treatment decision in the adjuvant setting should be guided by risk/benefit assessment in these patients: “Notably, the frequency and type of irAEs differed significantly in the adjuvant setting. This was particularly prominent in anti-PD1 treatment and in all grades of irAEs, but frequency of ≥grade 3 irAEs were similar between adjuvant and non-adjuvant setting. This observation is pertinent for risk/benefit assessment in adjuvant treatment, as many patients may already be cured by surgery alone. Furthermore, we observed a higher incidence of toxicities with long-term effects, or potentially fatal in the adjuvant setting with an increased frequency of myocarditis (6.7%), meningitis (2.2%), encephalitis (2.2%) and endocrinological toxicities (thyroiditis 17.8% and hypophysitis 13.3%). In fact, irAEs with long-term effects, or potentially fatal were more commonly observed than in the non-adjuvant setting and this risk should not be neglected in adjuvant treatment decision  in resected melanoma. The difference in frequency of irAEs might be attributed to the treatment exposure time and patient`s intrinsic factors, as well as immune status in stage III compared to advanced, unresectable melanoma”.

Comment 3: The authors should comment on why irAEs are correlated with increased treatment efficacy in the non-adjuvant setting but not in the adjuvant. What biological, clinical or statistical factors could have contributed to this difference and is this expected?

Response 3: Thank you very much for your comment. As also described in the text, our study did not confirm this superiority in risk of progression in the adjuvant setting. However, the small number of patients in this setting precludes from any statistical conclusion. The text has been revised accordingly: “In contrast to a recent report of the EORTC-1325/Keynote-054 study in patients with resected Stage III melanoma, where the presence of endocrine and skin toxicities in the pembrolizumab arm was correlated with improved RFS-rates, our study did not confirm this superiority in risk of progression in the adjuvant setting [17]. However, the small number of patients in this setting precludes from any statistical conclusion. Interestingly and besides the appearance of vitiligo-like depigmentation under interferons [27], no other reports with anti-PD1 or anti-CTLA4 monotherapy have been published so far confirming the prognostic value of irAE in the adjuvant setting either prospectively or retrospectively”.

Comment 4: Is there any association in irAEs between treatment naïve patients vs those who had previous immunotherapy vs those who had previous targeted therapy?

Response 4: We performed Fisher's exact test and Pearson's Chi-squared test to study any possible relationship between irAEs and previous treatment. In the adjuvant setting, results were non-significant (p>0.9). In the unresectable/non-adjuvant setting, previous targeted therapy (p=0.037) and immunotherapy (p=0.032) were associated with higher irAEs compared to treatment-naïve patients. Nevertheless, important co-variates, such as type of irAE and wash-out period have not been studied. Also, due to the low number of patients, the interpretation of these results should be made with caution. Due to the above mentioned reasons, we decided not to present these data in the current manuscript.

Adjuvant

Nonadjuvant

Previous treatment

Immunotherapy1

None1

p-value2

Immunotherapy1

None1

Targeted therapy1

p-value3

any AE

>0.9

0.003

None

0 (0%)

21 (33%)

3 (12%)

42 (34%)

14 (58%)

Yes

2 (100%)

42 (67%)

22 (88%)

81 (66%)

10 (42%)

1n (%)

2Fisher's exact test

3Pearson's Chi-squared test

Comment 5: Many previous studies have reported that anti-CTLA4 is more toxic compared to anti-PD1. However, in this study, the percentage of patients experiencing treatment related toxicities with anti-CTLA4 (59.2%) was similar to those with anti-PD1 (58.1%) in the non-adjuvant setting. Can the authors comment on this difference?

Response 5: Thank you for this observation. It is true that all-grade toxicities showed a similar frequency in both anti-CTLA4 and anti-PD1 in our cohort. However toxicities grade ≥3 occur more frequently in anti-CTLA4 (22.4%) and anti-CTLA4/anti-PD1 combination (58.8%), than in anti-PD1 (13.3%). Many previous studies either only reported on high-grade toxicities or there may have been a bias for underreporting low-grade toxicities. Nevertheless, we understand that anti-CTLA4 dosing might differ from the approved dose (in the adjuvant setting) and this might impact the observed toxicity rates. The text was updated accordingly, with this important observation: “While it is likely that such alterations in dosing and schedule do not impact efficacy or toxicity, the optimal treatment duration is not known and data from alternative dosing schedules from anti-CTLA4 should be thoroughly reviewed. In the adjuvant setting, ipilimumab is approved at a dose of 10mg/kg, but its use is limited due to high toxicity. As such, we applied adjuvant ipilimumab in the conventional dose of 3mg/kg 3-weekly, with significantly improved tolerability [28]. Although in the Checkmate-511 trial the induction dose of anti-CTLA4 does not seem to affect efficacy [29], in the Checkmate-915 trial, low-dose ipilimumab combined with nivolumab was not superior to nivolumab alone [30]”.

Comment 6: In their study, the authors concluded that application of steroids and immunomodulatory agents did not affect ICI efficacy. However, from previous studies (reference 31 and 32), this may be dependent on the duration of anti-PD-1 therapy. Did the authors consider this in their cohort (ie is the analysis stratified by duration of immune checkpoint inhibitors)?

Response 6: As mentioned in the discussion section, we support that early treatment or treatment in lower irAEs grades (e.g., grade 2) is preferable to prolonged treatment with high-dose corticosteroids. We underline this important observation in the discussion section: “…the impact of duration of the immunosuppressive treatment on disease outcome was not addressed in this study. In a secondary analysis of the EORTC-1325/Keynote-054 study, treatment effect of adjuvant pembrolizumab after 30 days of systemic steroid use appeared to be lower than without steroid use or by day 30 of systemic steroid use [17]”.

Comment 7: Figure 1D is a bit difficult to interpret, especially the x axis is not clear.

Response 7: We undertook some modifications to our figures, also according to the suggestion of Reviewer 1. We modified all figures so that readability is improved. We also omitted Figure 1E as well as figure 2E. Figure 2D has been modified and has been added separately, as Figure 2.

Comment 8: Figure 2C and 2D displayed as Waffle plots, although pretty to look at, is perhaps unnecessary and could be better summarised in a table format.

Response 8: Thank you for your comment. We tried to reduce the number of Figures and improve their quality in the revised version of our manuscript. We support the idea of a nicely designed Figure rather than a busy table to summarize our findings. As such, the co-authors decided to keep the present format of Figure 2C and 2D in the revised version of the manuscript.

Round 2

Reviewer 1 Report

Thank you to the authors for their careful consideration and response to all comments. I think the manuscript has been significantly improved by their changes.